# Radical Production with Pulsed Beams: Understanding the Transition to FLASH

**DOI:** 10.3390/ijms232113484

**Published:** 2022-11-03

**Authors:** Andrea Espinosa-Rodriguez, Daniel Sanchez-Parcerisa, Paula Ibáñez, Juan Antonio Vera-Sánchez, Alejandro Mazal, Luis Mario Fraile, José Manuel Udías

**Affiliations:** 1Grupo de Física Nuclear, EMFTEL & IPARCOS, Universidad Complutense de Madrid, CEI Moncloa, 28040 Madrid, Spain; 2Instituto de Investigación del Hospital Clínico San Carlos (IdISSC), Ciudad Universitaria, 28040 Madrid, Spain; 3Centro de Protonterapia Quironsalud, 28223 Madrid, Spain

**Keywords:** FLASH-RT, pulsed beams, radical recombination, ultra-high dose rate, GPU, chemical stage

## Abstract

Ultra-high dose rate (UHDR) irradiation regimes have the potential to spare normal tissue while keeping equivalent tumoricidal capacity than conventional dose rate radiotherapy (CONV-RT). This has been called the FLASH effect. In this work, we present a new simulation framework aiming to study the production of radical species in water and biological media under different irradiation patterns. The chemical stage (heterogeneous phase) is based on a nonlinear reaction-diffusion model, implemented in GPU. After the first 1 μs, no further radical diffusion is assumed, and radical evolution may be simulated over long periods of hundreds of seconds. Our approach was first validated against previous results in the literature and then employed to assess the influence of different temporal microstructures of dose deposition in the expected biological damage. The variation of the Normal Tissue Complication Probability (NTCP), assuming the model of Labarbe et al., where the integral of the peroxyl radical concentration over time (AUC-ROO) is taken as surrogate for biological damage, is presented for different intra-pulse dose rate and pulse frequency configurations, relevant in the clinical scenario. These simulations yield that overall, mean dose rate and the dose per pulse are the best predictors of biological effects at UHDR.

## 1. Introduction

Over the last few decades, a vast number of new radiotherapy modalities have emerged, including the use of different types of high energetic projectiles (such as protons or carbon ions), but also new procedures in dose delivery. Within this context, ultra-high dose rate irradiation (UHDR) or FLASH-RT has recently become a hot topic in radiation oncology. During a typical FLASH irradiation, all the dose is delivered in a single pulse or in a series of very short pulses with intra-pulse dose rates (~10^4^–10^9^ Gy/s) and mean dose rates of 40–100 Gy/s, so that the total delivery time is kept below 0.1 s [1,2]. The main benefit of this novel strategy is a considerable reduction of the radiation-induced toxicity in the normal tissue, maintaining a non-inferior antitumor effectivity than conventional radiotherapy (CONV-RT). This effect has been observed in vivo in mice, mini-pigs, zebrafish embryos and other animals [3,4,5] and constituted the basis of a single-patient, compressive treatment reported in 2018 [6]. Currently, two ongoing human clinical trials in bone and melanoma skin metastases are underway [7].

Although there is mounting evidence supporting the advantage of FLASH-RT over CONV-RT, its originating biological mechanism (and, in turn, the optimal beam parameters required to trigger it) have not been fully elucidated yet. Thorough research into finding these two key aspects is needed, since they are the cornerstone of the safe and optimal exploitation of this radiotherapy modality.

The topic of radiation chemistry in living systems has been extensively covered, both theoretically and experimentally [8,9]. Biological effects of radiation are ultimately due to two mechanisms: firstly, radiation can ionize the DNA or other molecules, causing direct damage in the cell, thus leading to the creation of organic (R•) radicals; secondly, indirect damage, derived from the ionization of water, is more frequent and generates several radiolytic products, including aqueous or hydrated electron (e^−^_aq_) and other reactive oxygen species (ROS), such as H_2_O_2_, O_2_^−^ or OH^−^, which can react with a variety of target molecules. For low-LET radiation, such as photons or clinical protons, these two effects account for approximately 1/3 and 2/3 of total cell damage [10], respectively. At FLASH dose rates, the concentration of generated free radicals is expected to exceed that of a conventional treatment. Eventually, this might change the kinetics of the different radiation-chemical reactions, and consequently, the cellular response to irradiation.

To date, there are at least three hypotheses which aim to explain the FLASH effect at the biological level [3,11,12]: (a) radiolytic oxygen depletion hypothesis, (b) radical-radical recombination hypothesis and (c) immune hypothesis.

### 1.1. Radiolytic Oxygen Depletion (ROD)

Oxygen has been described as a major player in the FLASH effect. Indeed, some studies report that FLASH effect is only observed at hypoxic concentrations below 12 mmHg (1.6%) [13] or 3.8 mmHg (0.5%) [14], while others [15] report that increasing oxygen concentration can partially reduce the protective effect of FLASH.

These observations could be explained by the so-called radiolytic oxygen depletion hypothesis, that is, that UHDR produces a transient radio-induced hypoxia in the tissue that limits indirect damage caused by radiation, causing the observed protective effect [1]. This hypothesis is based on the capacity of aqueous electrons (e_aq_^−^) and H• radicals to react rapidly with O_2_ via the e_aq_^−^ + O_2_ → O_2_^−^ and H• + O_2_ → HO_2_• reactions [16].

For initially hypoxic cells, which is commonly the case of tumors, this would result in a total oxygen depletion before capillary reoxygenation [17]. However, in normal tissue, the reduced amount of oxygen present at the end of the irradiation could also increase their radioresistance, thereby limiting the fixation of damage driven by oxygen and reducing the amount of reactive oxygen species produced at the end of the irradiation [15].

To confirm or refute this hypothesis, it is vital to measure adequately the rate of oxygen depletion inbiological media. Experiments conducted in the early 1960s and 1970s, both in water [18,19] and in cell cultures [20], yielded rates of 0.21–0.42 mmHg/Gy. These rates have been used by some authors [17] to present the ROD hypothesis as a plausible explanation for the FLASH effect. Other works, however, have reported a decreasing oxygen consumption rate with increasing dose rate [21,22], which would not support the ROD hypothesis. Furthermore, in both cases, the consumption rate was nearly independent of the starting oxygen level but increased linearly with the deposited dose.

Analytical models and Monte Carlo simulations can also shed some light on the role of oxygen in FLASH radiotherapy. Analytical models have suggested that ROD could be an explanation for the FLASH effect, but only for hypoxic cells [17,23] or in the physoxia range [24,25]. More detailed Monte Carlo simulations of oxygen depletion in the radiolysis of water at UHDR have reported that only high doses of radiation (>100 Gy) are able to produce any significant change in the oxygen enhancement ratio (OER) of relevant tissues [26,27]. Moreover, in these simulations, the radiolytic oxygen consumption decreased with increasing oxygen concentration and dose rate, in line with experimental data. In contrast, other recent computational studies still support the ROD hypothesis [28], some considering other variables, such as the distribution of capillaries in tissues [29].

### 1.2. Enhanced Radical-Radical Recombination

The theory of enhanced radical-radical recombination [30] implies that local high concentrations of radicals generated during the radiolysis of water at UHDR regimes favor their recombination, thereby reducing reactions with solvated oxygen. This same reasoning applies to reactions between organic compounds in cells, since the reaction rates of radical-radical recombination are proportional to the square root of the radical concentration [31]. This hypothesis would then explain the differencesbetween FLASH-RT and CONV-RT, while the competing reactions between different compounds would explain the observed differences between tumors and healthy tissues, as well as the aforementioned effect of oxygen concentration [32].

Recent experiments seem to support the radical recombination hypothesis [33]. A reduction in the yield of H_2_O_2_ has been observed under FLASH irradiation in water [15,34]. This decrease can be attributed to a reduction in the yield of the •OH radical, due to intertrack reactions in FLASH conditions [34,35], as it is the result of the reaction •OH + •OH → H_2_O_2_.

In biological media, due to the presence of several scavengers for reactive species, the half-life of many products of water radiolysis is very short, with reaction half-lives of 10^−8^–10^−9^ s [31,36]. Therefore, simulations in pure water might not be suitable for studying the FLASH effect, and biological species should be also considered. The role that other key biomolecules could play in the FLASH effect was first highlighted by Spitz et al. [10], discussing the importance of Fenton chemistry and peroxidation chain reactions in the differential fate of tumor and normal cells exposed to FLASH dose rates.

Following this idea, recently, Labarbe et al. [32] built a reaction-rate model including the most relevant reactive oxygen species (ROS) chemistry and antioxidant pathways in the cell: (a) Fenton reactions, (b) catalysis reactions with relevant enzymes, (c) reactions with several biomolecules, including lipids, proteins and DNA-generating radicals, (d) reactions with SH-containing compounds such as glutathione (GSH) or vitamin E (XSH) and (e) radical-radical recombination. Simulations based on this model suggest that UHDR irradiation increases the recombination of carbon-centered alkyl radicals (R•) (see Equation (1)), making them less prone to react with the cellular oxygen (Equation (2)), thereby reducing the overall cellular exposure to peroxyl radicals (ROO•). These molecules are known to be a major source of DNA and lipid damage in the cell, so reducing the exposure time to these reaction byproducts could be related to the FLASH protective effect.
R• + R• → R-R (1)
R• + O_2_ → ROO• (2)

### 1.3. Immune Hypothesis

Finally, several FLASH-RT studies have also reported differential immunological responses, including reduced activation of the TGF-b cascade in human lung fibroblasts [37] and mice [38] but also less pro-inflammatory cytokine levels [39]. More information about this topic can be found in [23,40]. In this case, it is suggested that the exposure of immune cells to radiation would be lower during FLASH-RT, due to the short time employed to deliver the prescribed dose, reducing the number of killed immune cells. However, further investigation regarding the effects of radiation in general on the immune response of the body is required.

### 1.4. Physical Beam Parameters

Based on experimental data, FLASH-RT is usually assumed when a threshold of 40 Gy/s on dose rate is reached and depending on the protective effect to be achieved. However, beyond the dose rate, there are several parameters (instantaneous dose rate, mean dose rate, duty cycle, total dose or treatment time) conforming a dose deposition pattern which may have an effect in the biological response. Ultra-high dose rate is generally defined simply by the certain minimum value of mean dose rate $\dot{D}$ (Gy/s) used to deliver the fraction of dose (*D*), which is in turn related to the total time of irradiation (*t_irr_*) by:(3)$$\dot{D}=\frac{D}{t_{irr}}$$

However, there is some evidence that, for a given mean dose rate during the fraction, a reduction in the number of pulses could lead to a reduction of radiation damage [41]. For the same average dose rate, intra-pulse dose rate can vary several orders of magnitude, depending on the frequency (or number of pulses, $N_{p}$) and the pulse width (t_pulse_) of the accelerator [42]:(4)$${\dot{D}}_{pulse}=\frac{D}{N_{p}\cdot t_{pulse}}$$

Thus, taking $\dot{D}$ as the sole responsible for the FLASH effect may be an oversimplification. Both experiments and simulations are required to further explore this subject and refine the set of parameters defining the FLASH regime.

### 1.5. Outline

The present paper aims to study the biological effects of different pulsation patterns on FLASH-RT, based on the radical recombination hypothesis at long time scales. For this purpose, we have developed a code to simulate the chemical and biological stages of water radiolysis at UHDR. The code is implemented in GPU and can be easily modified to include effects of oxygen or other relevant biological molecules. The AUC-ROO is selected as the relevant biochemical parameter for irradiation damage, and Normal Tissue Complication Probability (NTCP) values for a murine brain model and low LET radiation are calculated for different irradiation patterns, based on a literature model [32]. NTCP obtained from this model are evaluated for different intra-pulse dose rate and pulse frequency configurations. This framework is also applied to calculate radical production and expected biological outcomes for clinically relevant dose deposition patterns measured in a proton therapy facility and reported in previous works [43,44].

## 2. Results

### 2.1. Radical Production under Pulsed Irradiation

To study the influence of the beam temporal structures in the production of the peroxyl (ROO•) radical, simulations have been carried out considering identical dose fractions and mean dose rates ($\dot{D}$), with a continuous (no time structure) versus a pulsed beam. Following the prescription of Labarbe et al. [32], we have fitted our simulations to the experimental NTCP of Montay-Gruel et al. [45], who reported on the murine recognition ratio after 10 Gy irradiation at different mean dose rates ranging from CONV-RT (0.1 Gy/s) to FLASH-RT (>40 Gy/s) at a physiological oxygen concentration of 3.8% (50 μM). At the lowest dose rate of 0.1 Gy/s pulses of 1 µs width delivered at 10 Hz were used. Above this value, the pulse width and pulse repetition frequency were increased to 1.8 µs and 100 Hz, respectively [45]. During irradiation, ROO• radicals are formed from the reaction of R• and O_2_ (Reaction 2). Figure 1 illustrates the time evolution of the R• and ROO• species for three representative mean dose rates, ranging from CONV-RT (0.1 Gy/s) to FLASH-RT (100 and 500 Gy/s) simulated with a continuous and a pulsed beam irradiation, considering the above irradiation patterns.

The number of electron pulses in these simulations to deposit 10 Gy varies from 3 to 1001, with intra-pulse dose rates of 1.85·10^6^, 5.05·10^5^ and 9.99·10^3^ Gy/s. From these figures, it can be observed that at higher dose rates (100 and 500 Gy/s), the maximum concentration of the ROO• radical at the end of the irradiation ([ROO]*_max_*) is smaller when the dose is delivered in a series of short pulses. However, at 0.1 Gy/s, both irradiation modes give the same [ROO]*_max_* and no difference in the production of this species is observed at any time. The integral production or AUC (area under the curve) of the ROO• species is relevant to model the NTCP. It is given by:(5)$$\mathrm{AUC}=\int_{0}^{T}\left[\mathrm{ROO}•\right]\left(t\right)\cdot dt$$
where [ROO•] (t) is the molar concentration at each time. Table 1 list the [ROO•]*_max_* and AUC-ROO values obtained from these simulations.

The most relevant reactions contributing to the production of the ROO• radical are reactions numbered 46, 47, 48, 49, 51 and 53 in Table 6 of this work. Skipping any of these reactions in the simulation changes the AUC-ROO in more than 1%. However, the significance of reaction 48 changes at CONV and FLASH dose rates and only above 1000 Gy/s switching off this reaction changes the AUC-ROO in more than 1%. Similarly, the impact of reaction 51 on the AUC-ROO value is also influenced by the dose rate. In this case, increasing this variable raises the peak concentration [ROO•]*_max_* (see Table 1), and consequently, its self-recombination, which takes place via reaction 51.

These simulations can be extended to other pulse repetition frequencies at different mean dose rates (see Figure 2). The lower panel of the figure shows the ratio of AUC-ROO obtained with a pulsed beam with respect to the continuous irradiation. Error bars are not included in the figure, but each data point has an associated error of 6%, corresponding to one standard deviation. For the same physical beam parameters (i.e., pulse repetition frequency and pulse width), this ratio decreases with increasing mean dose rate. For mean dose rates below 10 Gy/s, there is no difference between the two irradiation modalities in the total exposure of the peroxyl radical. Above this value, the ratio between the two AUC-ROO values (pulsed and continuous) slightly decreases with increasing average dose rate and reducing pulse repetition frequency (number of pulses). At 1000 Gy/s delivered at 100 Hz, the maximum difference reached was about −3%.

### 2.2. Biological Effects of Pulsed Beam Irradiation

The previous section has shown that the time structure of the irradiation modifies the AUC-ROO and differences of up to 3% are observed (Figure 2) in terms of the irradiation pulse structure. To estimate the associated biological damage, we have employed the NTCP model proposed by Labarbe et al. [32]. The model relies on two parameters, γ and AUC_50_, that have been obtained from a fit to the experimental data of Montay et al. [45] (see Section 4.3) for C57BL/6J mice irradiated at CONV-RT and FLASH-RT dose rates with electron pulses. AUC_50_ represents the normalized AUC-ROO inducing a NTCP of 50% and γ is the slope of the response curve at AUC_50_. Both parameters are given in Table 2 and compared to those reported by Labarbe et al. [32]. In [32], a continuous beam and a fully homogeneous stage of radiolysis was employed in the simulation, while in this work, we do not assume the homogeneous assumption plus we introduce the pulse structure of the beam. This explains the minor differences observed between the values of the fitting parameters in both works.

Once we tuned our framework and verified that it yields results consistent with [32], we set out to investigate in detail the influence of the accelerator duty cycle on the production of the ROO• radical and in the corresponding biological damage. We used fixed values of total dose of 10 Gy and an oxygen concentration of 3.8% in order to focus solely on the effects of the time structure of the beam.

#### 2.2.1. Effect of the Number of Pulses

Setting a pulse size of 1.8 μs, we study different numbers of pulses, i.e., different intra-pulse dose rates and duty cycle/time between pulses/frequency. The total time of the irradiation is given by the prescribed $\dot{D}$. Figure 3 shows the temporal evolution of the ROO• radical for mean dose rates of 5 Gy/s (a) and 100 Gy/s (b), from the extreme case in which just two pulses are employed up to the semi-continuous mode of irradiation (${\dot{D}}_{pulse}\;$≈$\;\dot{D}$). Reducing the number of pulses has a sizeable effect in lowering the production of radical and the [ROO•]*_max_* at 100 Gy/s.

Table 3 lists the AUC-ROO production for the same average dose rate irradiation and different intra-pulse dose rates. The difference increased with the number of pulses and the average dose rate, with relative differences of 2.8% at 5 Gy/s and 3.4% at 100 Gy/s between the two-pulse irradiation and the semi-continuous irradiation.

Figure 4 shows the resulting NTCP obtained with the fitting parameters listed in Table 2, as a function of the dose per pulse and the corresponding intra-pulse dose rate (Gy/s). Four different mean dose rates, 5, 10, 100 and 1000 Gy/s, have been investigated here. A decrease in the biological damage probability is observed when the dose per pulse is increased above ~0.2 Gy (equivalent to intra-pulse dose rates of about 10^5^ Gy/s). As in the proposed model, the NTCP changes very fast with the AUC in the threshold regions, at FLASH-RT dose rates (at 100 Gy/s and above), a relatively small difference in AUC-ROO between pulsed and continuous regimes can shift the NTCP from 0.4–0.6 to nearly 0.

#### 2.2.2. Effect of the Pulse Width

The effect of the pulse width is now analyzed by comparing irradiations at different mean dose rates while keeping the pulse repetition frequency at 100 Hz. Figure 5 shows the time evolution of the ROO• radical when a fraction of 10 Gy is delivered keeping the same average dose rate (a) 5 Gy/s and (b) 100 Gy/s with three different pulse widths. The semi-continuous mode represents the case in which the pulses are superimposed.

For the same average dose rate and number of pulses, wider pulses give slightly larger concentrations of the ROO• radical (see Table 4). However, the effect of the pulse width is much less significant than that of the number of pulses and the relative change of the AUC-ROO in the semi-continuous mode with respect to the simulations with 1 μs pulses is below 1% for 100 Gy/s, while no difference in radical production is observed for pulses of different widths at 5 Gy/s.

The effect in the biological response is illustrated in Figure 6, where the NTCP is plotted against the pulse width for mean average dose rates of 5, 10, 100 and 1000 Gy/s.

Changing the pulse width can shift the NTCP from 0.2 to 0.4 at 100 Gy/s, whereas at larger UHDR, such as 1000 Gy/s, the biological protection is preserved for all the pulse widths (and intra-pulse dose rates) considered.

### 2.3. Dose-Response Curves with Different Clinical Beams

The temporal microstructure of the proton beam delivered by the Proteus One accelerator placed at the Quiron Protontherapy Center in Madrid (Madrid, Spain) has been characterized in a set of previous measurements [43,44]. The accelerator delivers 10 μs pulses each 1 ms (1 kHz).

To study the predictions of the NTCP model assumed here and in ref. [32], dose-response curves at CONV-RT (0.1 Gy/s) and FLASH-RT (100 Gy/s) have been calculated considering the time structure of the Proteus One accelerator and the physical parameters reported in the work of Montay et al. [45] for the Oriatron 6e (see Figure 7). Additionally, NTCP predictions based on a continuous beam are also included. All of them are displayed for two different levels of oxygenation. A FLASH modifying factor of 1.49 and 1.56, defined as the ratio of the FLASH dose to the conventional dose rate dose [46], is obtained in both cases. This value is in good agreement with previous experimental work that reported values in the range of 1.36–1.58 [5,47]. At FLASH dose rates, the temporal microstructure of the accelerator changes the D_50_ by just 0.2–0.3 Gy.

## 3. Discussion

Full exploitation of the UHDR healthy tissue protective effect is being regarded as the next groundbreaking innovation in radiotherapy. However, the scarcity of experimental data together with the small number of devices capable of conducting new experiments in this area makes it difficult to disentangle the role of the different biological factors and the correct definition of the physical variables implicated in the sparing FLASH effect. Furthermore, to assess the possible effect of physical and biochemical components of irradiations, dedicated computational tools are necessary.

In the present paper, we have reported the basis of a Monte Carlo-based GPU tool aimed at performing simulations of the chemical (heterogeneous) and biological (homogeneous) stages of the radiolysis of water and cellular medium, under different dose rates (CONV-RT and FLASH-RT) and beam time structures. The initial physical and pre-chemical stages of the simulation of the radiolysis of water are based on the Monte Carlo tool TOPAS-nBio [48], to introduce location and amount of ionization events. In the organic media, we have reproduced the work of Labarbe et al. [32], but without the simplifying assumptions they employed, as instead, our predictions included the heterogeneous stage in detail with a numerical simulation. Furthermore, if we assume the same prescription of [32] for the correlation of NTCP with AUC-ROO, we obtain a very similar fit to the experimental data for NTCP of Montay et al. [45], with the differences in the treatment of the heterogenous stage reflecting in a readjustment of the fitting parameters of about −15% and 22% for the AUC_50_ and γ, respectively. This can be explained as follows.

We have considered that all direct DNA damage take place during the initial physical events of radiation [49]. Therefore, simulation of the non-homogeneous chemical stage, up to the 1 μs, increases the G-value of the R• radical reported at this time due to the ongoing reactions, thus giving a larger AUC-ROO value at the end of the irradiation. This variation is more relevant at CONV dose rates because in this regime the reaction of the R• radical with oxygen (reaction 46) is favored, and there is a linear correlation between the fraction of dose delivered and the AUC-ROO calculated in the simulation. This changes the normalization of the AUC values to the maximum AUC-ROO, which is simulated at 30 Gy at 0.03 Gy/s [32], and therefore, it is larger in this work. Besides this, the differences in the AUC_50_ and γ can also be explained by the decrease in the AUC-ROO derived from the simulation of the time structure of the beam, as shown in Figure 2.

Our detailed simulation tool makes it possible to study the impact of the temporal microstructure of the irradiations in the AUC-ROO and thus NTCP. In this study, we have investigated the possible effect in the biological response of variations of the beam time structure, for the same mean dose rate.

We have found that the temporal microstructure of the irradiation (pulse width, number of pulses) has an effect on the time evolution and integral production of the ROO• radical only for high average dose rates, precisely near the customarily accepted onset of the FLASH regime (≥10 Gy/s), as can be observed in Figure 2. Even when the effect of the irradiation pattern in the AUC-ROO is of just a few percent, as this happens at the region of very high slope for the NTCP response, the corresponding change in NTCP can be very noticeable. Indeed, for a given average dose rate and total dose in the fraction, the pulse repetition frequency or the dose per pulse may have a large impact on the treatment outcome in terms of NTCP, provided the dose per fraction and dose rate are within the FLASH-RT region. Thus, while changing the dose per pulse from 5 to 0.1 Gy decreased the integral production of the peroxyl radical by just 4%, the induced change in NTCP is very large. It is worth mentioning that previously available experimental data also showed a reduction in the radioprotective effect of UHDR when the dose fraction was delivered with a large number of pulses of the same width [41]. Similarly, recent Monte Carlo simulations of the radiolytic oxygen consumption in FLASH-RT have found a relatively stronger influence of the dose per pulse, while the intra-pulse dose rate had no effect, for the same average dose rate [50]. Other experimental studies point towards the mean dose rate alone as the best predictor of the manifestation of FLASH effect [51,52], which could be consistent with the small radical variations reported in this work. Further studies are then needed to elaborate more predictions of the biological damage in terms of radical production.

Our simulations show that, for fixed dose and average dose rate, the steady (final) state of the ROO• radical depends on the pulse repetition frequency and accordingly on the total dose deposited in each pulse. Therefore, increasing the number of pulses reduces the equilibrium concentration after each pulse. This is well illustrated in Figure 4. Thus, the effect of the pulse width is less noticeable on the variation in the [ROO•]*_max_*, as this variable does not modify the equilibrium state nor the transient concentrations of competing radicals, providing that the same number of pulses is employed. One can find a parallelism with the behavior of ionization chambers, where ion recombination plays an important role, being strongly dependent on the dose per pulse rather than the intra-pulse dose rate [42].

The study presented here presents certain limitations, some of them already mentioned in Labarbe et al. [32], which arise from the poor knowledge of the reaction rates and diffusion constants that govern a real biological environment, as well as the differences in these parameters between normal and tumor cells. Although the more relevant biochemical pathways triggered after an ionization event are included in the simulations, the influence of other molecules should also be investigated. For instance, another important source of cell damage and oxidative stress during irradiation arises from the production of reactive nitrogen species (RNS). In the presence of oxygen, RNS are produced from the ROS generated after the ionization event. For instance, in the presence of nitric oxide (NO), O_2_^−^ reacts very quickly yielding peroxynitrite anion (ONOO^−^) [53]:O_2_^−^ + NO → ONOO^−^
(6)

The rate constant for this reaction is close to the diffusion-controlled limit (k~1.9 × 10^11^), and thus, it is larger than the decomposition reaction of O_2_ by the superoxide dismutase [54]. Peroxynitrite is in equilibrium with peroxynitrous acid (ONOOH) and both species show a great reactivity towards a wide range of biological targets, including lipids, thiols, sulfides, amino acid residues in proteins and DNA bases [55].

For the simulation of the physical and pre-chemical stages of the radiolysis, the default cross-sections, branching ratios and dissociative schemes were used as input data for TOPAS-nBio. All of them are defined in a water medium, which is used as a surrogate of the biological material. Changing these parameters does not result in large variations of the production yields of inorganic radicals [56], which are also modulated by the high scavenging activity of the organic media. However, in a biological cell, the atomic composition and density also varies across the different cellular constituents, such as the cell nucleus, the cytoplasm, mitochondria or lysosomes. Ionization cross-sections and the sensitivity of these targets to the ionizing particles should also be revised to obtain more precise predictions on the production yields of carbon-based radicals [57,58].

Simulation of the heterogeneous chemical stage of the radiolysis of water often involves the solution of the diffusion or Smoluchowski equation. For this purpose, several stochastic models have been developed, such as the step-by-step (SBS) or the independent time reaction (IRT) methods, which are implemented in the Geant4-DNA and TOPAS-nBio simulation tools [59]. The SBS method is computationally more expensive but allows for accurate simulations of different biological molecules. The IRT method is more efficient (about two orders of magnitude) but it poses a problem to reactions of radicals with static molecules. In both cases, the position of each radical species or the distance between reactant pairs is scored during all the simulation. Recently, in silico studies have been performed with both methods to investigate the effects of FLASH dose rates on the heterogeneous stage of the radiolysis of water [35] but also the indirect damage on relevant biomolecules, such as DNA [59].

Mendez et al. [35] performed a series of simulations at CONV and FLASH dose rates to evaluate the intertrack effects on the primary yields of the radiolysis products, owing to the high local radical concentrations. Their work is based on the TOPAS-nBio IRT method and they reported a variation on the G-values of the e^−^_aq_ and OH species to a maximum of 20%. Tian et al. [59] have also simulated the biological damage on a DNA chromatin fiber, using a different implementation of the IRT method.

In contrast, the approach employed in this work does not explicitly follow each radical species produced in the radiolysis of water. In a real biological medium, most of these products are scavenged fast and the exact position of the different cellular modulation systems, such as metals, enzymes or antioxidants, is not known. Therefore, one option is to assume that all these reactants are homogeneously distributed in the different regions of the target, where they are treated as a continuum. Furthermore, by doing this, the reaction set can be easily extended to include other species of radiomodifiers or nanoparticles at different concentrations and spatial distributions. The effect of spatial heterogeneities, such as cellular boundaries, can also be incorporated into the simulations.

More recently, Mendez et al. [60] conducted a simulation study on the temperature dependance of single (SSB) and double (DDB) indirect DNA strand breaks. Their results were compared to experimental data, showing a linear relation between the yield of SSB and DDB with temperature and dose. In this work, the reaction rates and diffusion coefficients are reported at ambient temperature and have been assumed constant during the calculations. However, the temperature effects on the water radiolysis processes and for the reactions between the chemical species and the target molecules should also be addressed to analyze these effects in the radiobiological outcome of the model. Studying the influence of these factors goes beyond the focus of the present study but future work will include these scenarios, as well as further optimization of the code and its GPU implementation.

## 4. Materials and Methods

### 4.1. Simulation of the Chemical Stage at Different Dose Rates

The non-homogeneous sequence of events which takes place during the radiolysis of water can be divided into three different time scales: the physical (~10^−15^ s), pre-chemical (~10^−15^−10^−12^) and chemical stage (~10^−12^–10^−6^). During the physical and pre-chemical stages, the passage of radiation essentially produces the excitation and ionization of the water molecules, followed by the relaxation of the media, giving some stable molecules and free radical species, such as H^+^, •OH and e^−^_aq_. In the chemical stage, the different species start to react and diffuse away from the original track in a non-homogeneous way. More information about the details of these different stages can be found in [61,62] The biological stage is considered to start right after these first stages and includes the cell response to the different products of the irradiation.

In this work, we have chosen the TOPAS-nBio tool (TOPAS-nBio v1.0) [48,63] for the simulation of the physical and pre-chemical stages. TOPAS-nBio is an extension of the TOPAS MC tool aimed to perform detailed Monte Carlo simulations at the micro and nano scales. The goal of this tool is to provide a user-friendly interface to understand radiobiological effects at the (sub)cellular level, intended for professionals in the field without programming knowledge. Simulation parameters are specified by simply text parameter files and a set of pre-defined cell geometries, scorers and chemical reactions can be used for the simulations. TOPAS-nBio is based on the Geant4-DNA package [64], which is an extension of the GEANT4 toolkit that allows to perform Monte Carlo track-structure simulation of particles in liquid water down to very low energies (~eV) on the nanometer scale, as well as the production and chemical reactions of different radiolytic products.

The initial physical interactions of the radiolysis of water were simulated with the default physics lists of TOPAS-nBio, which are included in the TsEmDNAPhysics module [63]. This module is composed of several Geant4-DNA models which are used to describe different physical electronics processes, including inelastic (G4DNABornExcitationModel and G4DNABornIonisationModel) and elastic scattering (G4DNACPA100ElasticMode), vibrational excitation (G4DNASancheExcitationModel) and Dissociative attachment (G4DNAMeltonAttachmentModel) [65].

Chemical interactions during the pre-chemical stage were also simulated with the TsEmDNAChemistry model, which includes revised water dissociation schemes and branching ratios from the G4EmDNAChemistry model [63].

To simulate the initial species at 1 ps, a monoenergetic source of 1 MeV electrons was uniformly distributed at the edge of a homogeneous 5 × 5 × 5 μm^3^ cubic water phantom (see Figure 8). The position of each radical species and the total dose in the volume at the end of the pre-chemical stage were computed and stored for each individual simulated history, using the particle n-tuple and dose scorers provided by TOPAS-nBio. This scorer contains the information about the molecule ID, spatial coordinates (x,y,z) and track of each radical species at the time specified by the user. This was done for a total of 10^3^ different histories, with different random number seeds.

For the simulation of the chemical stage, these two outputs obtained from TOPAS-nBio were then used as input data in an analytical GPU code, designed to include the beam time structure and dose rate during this stage. The time structure was modeled by varying the number of histories to deliver a given fraction of dose in a pulse of width t_pulse_. The arrival time of each history within each pulse has been sampled from a uniform distribution [35]. For conventional dose rates, when few particles arrive in the volume, the simulation can be reduced to the simplest case in which just one history at a time is followed. In this case, the simulation was repeated with 100 histories and the results averaged.

The simulation volume for the chemical non-homogeneous stage consisted in oxygenated water modelled with a 3D array composed of 50 × 50 × 50 cells of 100 nm size, as depicted in Figure 8. Carbon-centered radicals (RH) and other relevant biomolecules, such as glutathione (GSH), lipids, iron or enzymes, used to model the cell environment, are also assumed to be homogeneously distributed in the volume with a constant concentration [32]. Direct ionization of these carbon-based molecules to yield carbon-centered alkyl radicals (R•) was modeled considering the spatial distribution of the initial energy depositions onto the geometrical volume and a radiolytic yield for direct DNA ionization of 0.569 mol/100 eV, which includes the total direct strand breaks [32]. The initial chemical species simulated in TOPAS-nBio were distributed into this volume, according to its scored spatial position. The concentration of this radiolytic species within each cell is simply given by c^s^_i,j,k_ = N_i,j,k_/V_cell_, with V_cell_ being the volume of each 3D cell considered in the simulation (V_cell_ = 0.1 × 0.1 × 0.1 μm^3^) and N the number of molecules produced in the beam direction within that volume.

The time evolution of each species is described by a set of coupled nonlinear ordinary differential equations, illustrated in Figure 8. D_s_ is the diffusion coefficient of each chemical species and K is the second order rate constant, considering the set of reactions in which the given species is produced (+) or consumed (−). The indexes i, j and k represent the position in the water target. Both equations were solved numerically in parallel using an Nvidia GeForce GTX 1080 Ti GPU.

The chemical reactions covered by TOPAS-nBio were extended, and the full set is included in Table 5. All of them correspond to a fixed temperature of 25 °C. The corresponding reaction rates were taken from the work of Elliot et al. [66]. For the simulations in biological media, the reactions with biomolecules considered in the work of Labarbe et al. [32] and their concentration were also included in the calculations (Table 6). Diffusion coefficients were set to the same values employed in TOPAS-nBio, which are listed in Table 7, and periodic boundary conditions were applied. Oxygen was kept constant at the borders, with the initial value for the partial pressure being considered. The partial pressure can be simply related to the oxygen concentration in the volume applying Henry’s law (p = c·H), where H is the Henry’s constant, which at T = 25 °C is H = 1.71 × 10^−6^ mol/mmHg.

### 4.2. Simulation of the Biological Stage

The chemistry stage finishes at 1 µs and from this moment all the species are assumed to be homogeneously distributed in the medium. We can dispose of the spatial distribution and the molecular diffusion and study the time evolution of the different reactants at the rates defined by the chemical yields and the dose rate. For irradiation times ≤ 1 µs, this stage just comprises the chemical evolution of the species. Otherwise, the production term is added to the set of coupled differential kinetic Equations:(7)$$P_{s\;}\left(M/s\right)=\dot{D}\left(Gy/s\right)\cdot G_{s}\cdot \frac{1}{N_{av}f_{c}}\cdot \rho$$
where $\dot{D}$ (Gy/s) is the corresponding dose rate, $G_{s}$ the chemical yield of the species s (molecules/100 eV), $N_{av}$ the Avogadro number, ρ the density of water in Kg/L and f_c_ = 1.602 10^−17^ (J/100 eV), a conversion factor to change units of 100 eV into J. This production term is added to the set of equations until the total dose has been delivered. The chemical reaction of the different species is continued for longer times, typically in the order of 10^2^ s, covering the first processes taking place during the biochemical and biological phases of radiation [67]. A workflow of the full simulation process is depicted in Figure 9.

### 4.3. Modeling of Normal Tissue Complication Probability

For this work, the expected amount of biological damage is based on a normal-tissue complication probability (NTCP) model fitted to experimental data [45]. The model depends on the total cell exposure to the ROO• radical [32] and is based on a sigmoid function with two fitting parameters, γ and AUC_50_:(8)$$NTCP_{model}\left(D,\frac{dD}{dt}\right)=\frac{1}{1+e^{-\gamma \left(AUC_{norm}\left(D,\;\frac{dD}{dt}\right)-AUC_{50}\right)}}$$
where *AUC_norm_* is the normalized AUC-ROO for a given dose and dose rate. All the results have been normalized following the same criteria of Labarbe et al. [32]. Then, *AUC_high_* in all cases takes the same AUC value for continuous and pulsed modes. This corresponds to the assumption that at low dose rates, the number of pulses is expected to be very large for the frequency and pulse width considered so that the production of ROO• does not differ between both irradiations, as has been shown in Figure 1. For the highest dose rate and lowest dose, the minimum AUC value, denoted as *AUC_low_*, corresponds to the case for which at least two pulses of radiation are employed in the simulation to deliver a total dose of 1 Gy. This is equivalent to an average dose rate of $\dot{D}$= 100 Gy/s, for a pulse repetition frequency of 100 Hz.

Figure 10 shows the sigmoid fit of the NTCP to the experimental data (Equation (8)), considering the production of the ROO• radical in the pulsed beam simulations. This fit has been compared to the results reported by Labarbe et al. [32] with continuous irradiation and without considering the heterogeneous chemical stage of the irradiation. Both fits are very similar in predicting the biological response as a function of the dose rate to the experimental data.

### 4.4. Sensitivity Study

The reaction rate coefficients and the concentration of the different molecular compounds used in this work might be different in a real cell environment. A nuisance analysis was performed to quantify the influence of these variations on the ACU-ROO value. Then, all the reaction rates and the concentrations listed in Table 5 and Table 6 were resampled from a random normal distribution, within a ±50% range for a total of 200 independent simulations. From each simulation, the AUC-ROO has been computed. Following this procedure, a variation of AUC-ROO of ± 6% (1 sigma) is found, very similar for high and low dose-rate simulations. This uncertainty is included in all the data listed in this work.

## 5. Conclusions

In this work, a computer code implemented on a GPU to assess the impact of the time microstructure of the dose deposition on the expected amount of biological damage has been developed. The use of a GPU allows for detailed simulations, including the heterogeneous stage, and overcomes simplifications employed in some previous works. Time microstructures of the irradiation can also be contemplated. The transition to the so-called FLASH regime for several intra-pulse dose rates and pulse frequency configurations has been investigated, using a NTCP model based on AUC-ROO. The model was fitted to experimental data and find similar parameters and good agreement with Labarbe et al. [32]. The small differences in the parameters of the fit with the previous work can be attributed to the simulation of the heterogeneous chemical phase in this work, and the consideration of the pulsed structure of the irradiation. Within this model, there is significant dependence of the NTCP with the time structure of the irradiation, particularly at the onset of FLASH-RT, where NTCP changes of up to 60% are observed. These results indicate that a detailed simulation as the one introduced here is compulsory to understand the FLASH effect. Biological damage should be evaluated over a broad range of beam structures. In the near future, these simulations will be extended by the addition of new biologically significant reactions and compared to different experimental data.

## Figures and Tables

**Figure 1 ijms-23-13484-f001:**
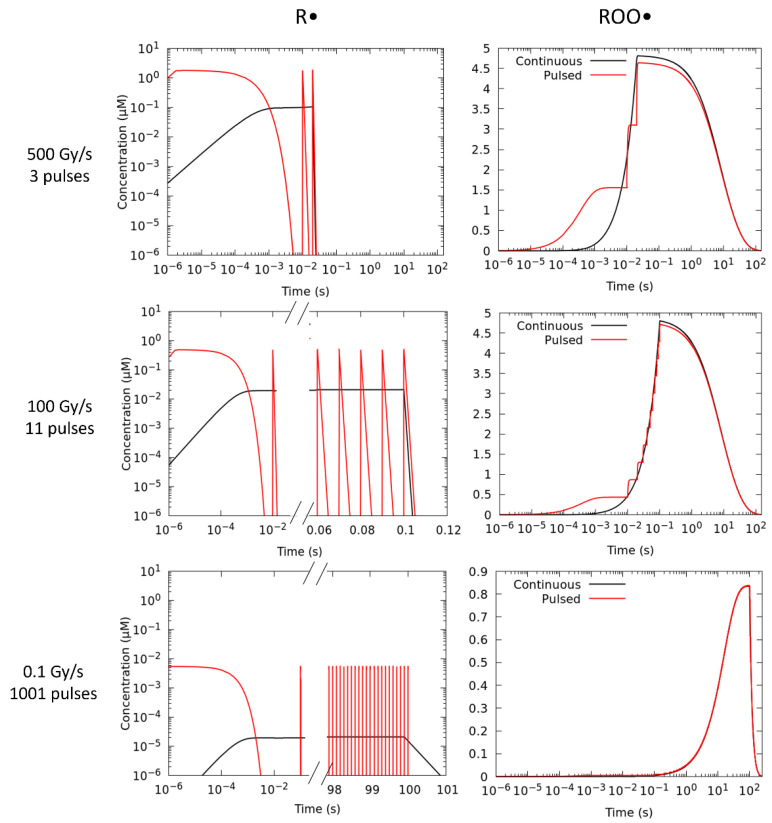
Production of the R• and ROO• radicals in a 10 Gy fraction simulated with a continuous (black) and pulsed (red) 1 MeV electron irradiations for three different mean dose rates, including both FLASH–RT: 500 Gy/s (top panels), 100 Gy/s (middle panels) and CONV–RT: 0.1 Gy/s (bottom panels) regimes. The number of pulses used to deliver the prescribed dose was 3, 11 and 1001.

**Figure 2 ijms-23-13484-f002:**
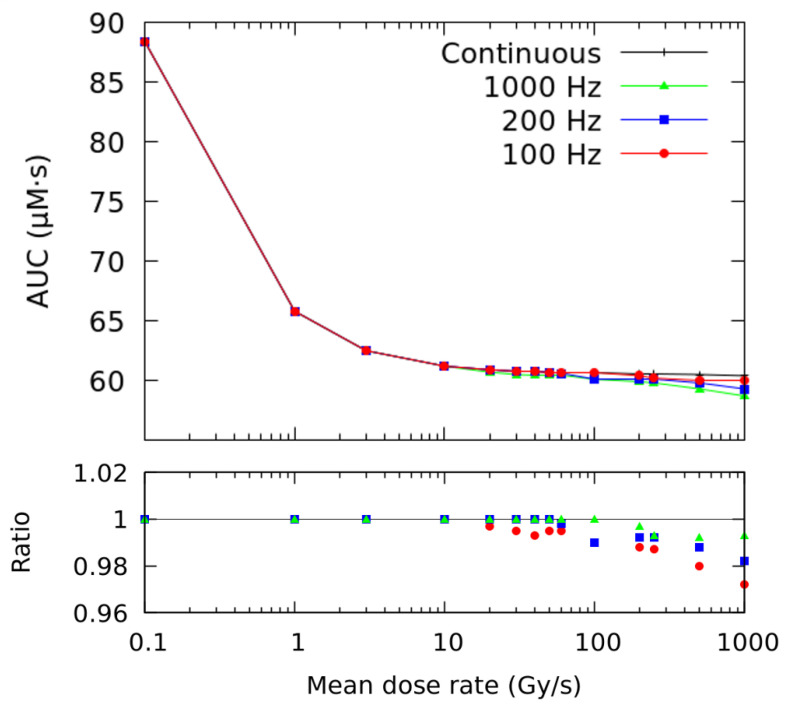
AUC (Area Under the Curve) of the ROO• radical as a function of the mean dose rate for continuous (black) and pulsed irradiations at different pulse repetition frequencies. The total dose was 10 Gy and the pulse width 1.8 μs. The ratio for different pulsed beams with respect to the continuous irradiation is shown in the lower panel (note: error bars are not included in the figure; however, each data point has an uncertainty of 6%, which represents one standard deviation).

**Figure 3 ijms-23-13484-f003:**
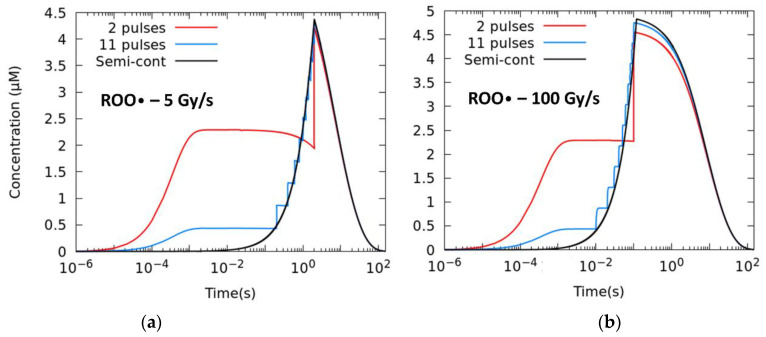
Time evolution of the ROO• radical for different degrees of pulsatility. The simulations correspond to a total dose of 10 Gy with 1.8 μs width pulses and mean dose rates of (**a**) 5 Gy/s and (**b**) 100 Gy/s.

**Figure 4 ijms-23-13484-f004:**
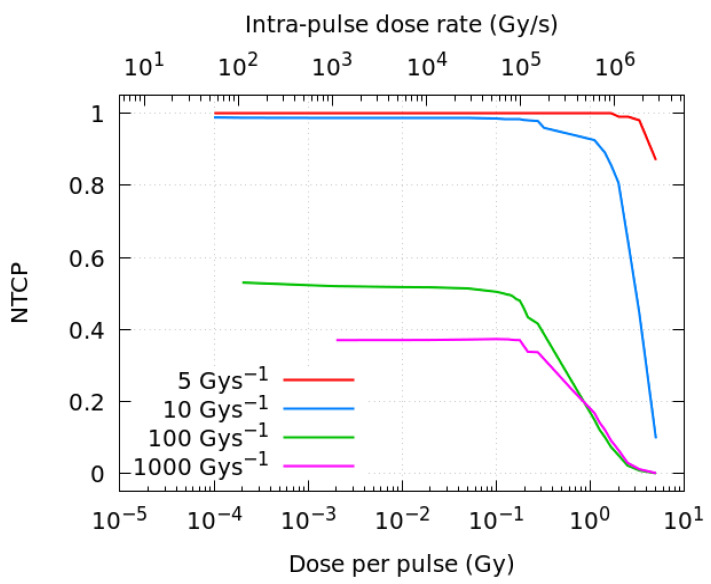
NTCP as a function of the dose per pulse and the intra-pulse dose rate for different average dose rates of radiation. All the simulations correspond to a total dose of 10 Gy.

**Figure 5 ijms-23-13484-f005:**
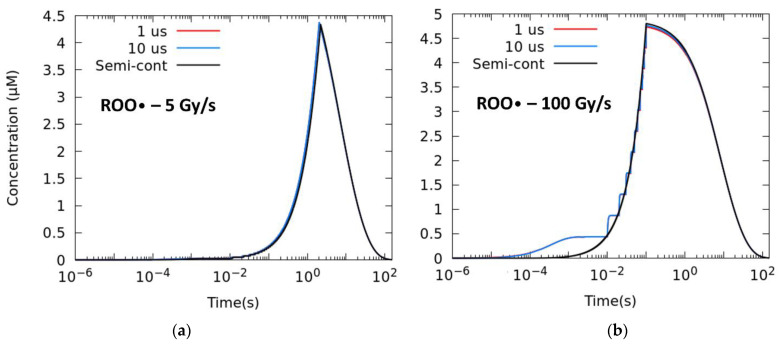
Time evolution of the ROO• radical at (**a**) 5 Gy/s and (**b**) 100 Gy/s with a frequency of 100 Hz and pulses of different widths. The simulations correspond to a total dose of 10 Gy.

**Figure 6 ijms-23-13484-f006:**
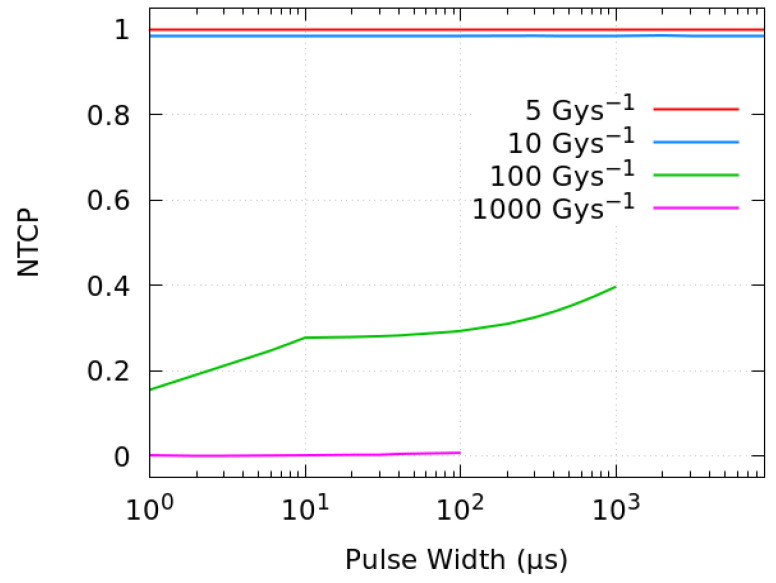
NTCP as a function of the pulse width for different mean dose rates. The simulations correspond to a total dose of 10 Gy delivered at 100 Hz.

**Figure 7 ijms-23-13484-f007:**
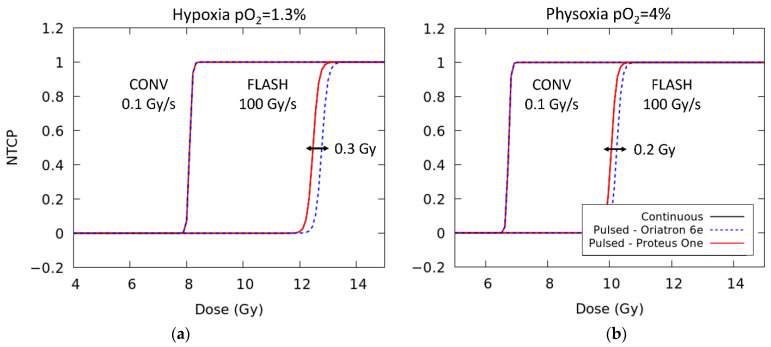
Dose-response curves at either conventional dose rate (0.1 Gy/s) or FLASH dose rates (100 Gy/s) for two different levels of oxygenation: (**a**) 1.3% oxygen and (**b**) 4% oxygen. For each dose rate, the results are displayed considering the temporal microstructure of two clinical accelerators (Proteus One [44] and Oriatron 6e [45]) and a continuous beam irradiation.

**Figure 8 ijms-23-13484-f008:**
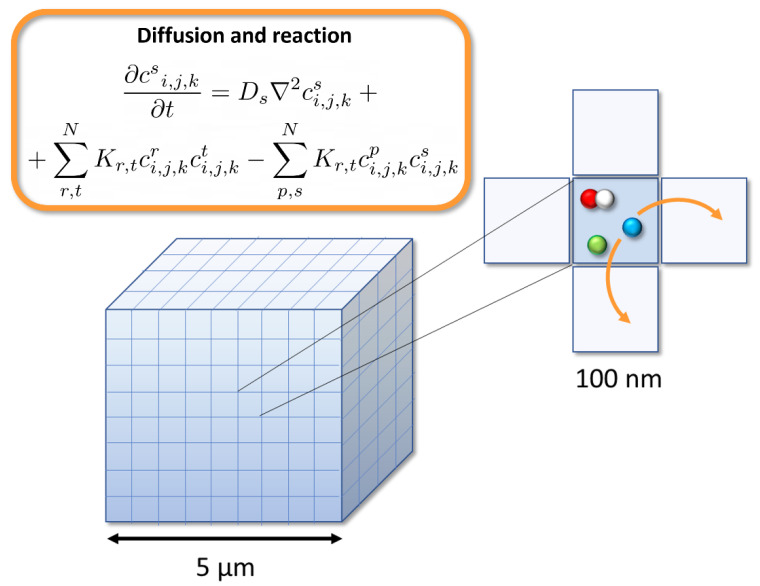
3D lattice employed in the simulations and diffusion and reaction scheme between adjacent cells.

**Figure 9 ijms-23-13484-f009:**
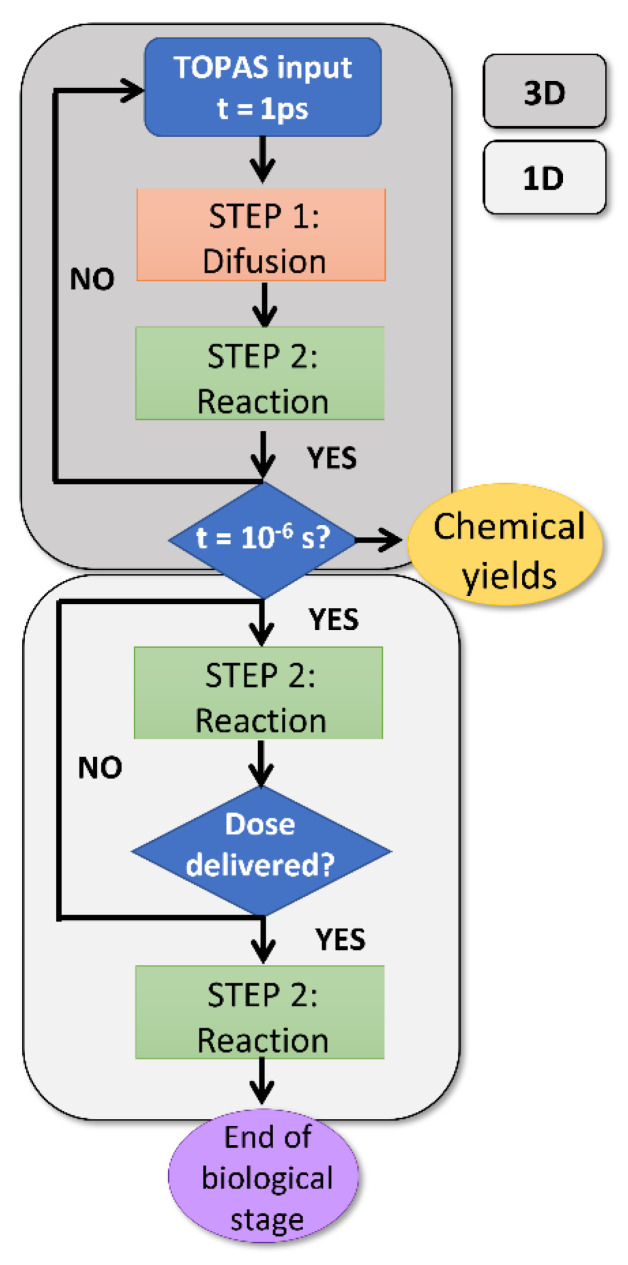
Workflow used for the simulation of the chemical and biological stages of the radiolysis.

**Figure 10 ijms-23-13484-f010:**
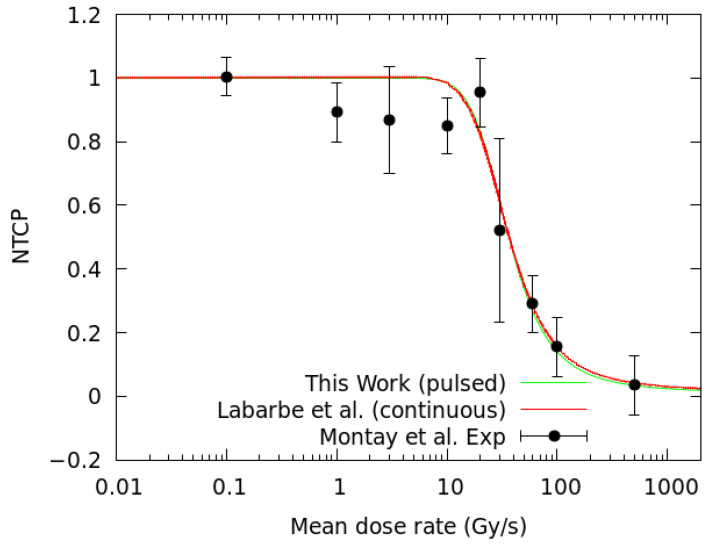
NTCP logistic fit to the experimental data of Montay et al. [45]) for a pulsed (green) beam irradiation using the same physical parameters (frequency and pulse width) of the experiment. The red line represents the same fit obtained in the previous work of Labarbe et al. [32] with a continuous beam (radiation with no time structure).

**Table 1 ijms-23-13484-t001:** Maximum concentration and integral production of the peroxyl radical obtained with a continuous and a pulsed beam for three different dose rates employed in the work of Montay et al. [45] to irradiate mice. Reported data uncertainties correspond to one standard deviation.

Mean Dose Rate (Gy/s)	[ROO•]_max_ (μM)		AUC-ROO (μM · s)	
	Pulsed	Continuous	Pulsed	Continuous
0.1	0.84 (5)	0.84 (5)	89 (5)	88 (5)
100	4.7 (3)	4.8 (3)	60 (4)	61 (4)
500	4.6 (3)	4.8 (3)	59 (4)	61 (4)

**Table 2 ijms-23-13484-t002:** Fitting parameters for Equation (8) and their respective errors, corresponding to the fits shown in Figure 10 (Section 4.3).

Parameter	Labarbe et al. [32]	This Work
γ	1287	1000 (300)
AUC_50_	0.199	0.1682 (3)

**Table 3 ijms-23-13484-t003:** Integral production of the ROO• radical for mean dose rates of 5 and 100 Gy/s and corresponding irradiation parameters used in the simulations of different pulse repetition frequencies, keeping the pulse width to 1.8 μs. Reported data uncertainties correspond to one standard deviation.

		5 Gy/s		100 Gy/s	
No. Pulses	Intra-Pulse Dose Rate (Gy/s)	Frequency (Hz)	AUC-ROO(µM · s)	Frequency (Hz)	AUC-ROO (µM · s)
2	2.78 × 10^6^	0.5	60 (4)	10	59 (3)
6	9.26 × 10^5^	2.5	61 (4)	50	60 (3)
11	5.05 × 10^5^	5	61 (4)	100	60 (3)
21	2.64 × 10^5^	10	61 (4)	200	60 (3)
51	1.09 × 10^5^	25	61 (4)	500	60 (3)
101	5.50 × 10^4^	50	62 (4)	1000	61 (3)
Semi-continuous	1.11 × 10^2^	2.50 10^4^	62 (4)	5.00 10^5^	61 (3)

**Table 4 ijms-23-13484-t004:** Integral production of the ROO• radical for mean dose rates of 5 and 100 Gy/s and corresponding irradiation parameters used in the simulations of different pulse widths, keeping the pulse repetition frequency at 100 Hz. Reported data uncertainties correspond to one standard deviation.

	5 Gy/s		100 Gy/s	
Pulse Width (μs)	Intra-Pulse Dose Rate (Gy/s)	AUC-ROO (µM · s)	Intra-Pulse Dose Rate (Gy/s)	AUC-ROO (µM·s)
1	4.98 × 10^4^	62 (4)	9.09 × 10^5^	60 (3)
6	8.29 × 10^3^	62 (4)	1.52 × 10^5^	60 (3)
10	4.98 × 10^3^	62 (4)	9.09 × 10^4^	60 (3)
65	7.65 × 10^2^	62 (4)	1.40 × 10^4^	60 (3)
100	4.98 × 10^2^	62 (4)	9.09 × 10^3^	60 (3)
1000	4.98 × 10^1^	62 (4)	9.09 × 10^2^	61 (3)
Semi-continuous	5.53 × 10^0^	62 (4)	1.01 × 10^2^	61 (3)

**Table 5 ijms-23-13484-t005:** Reaction set used in the simulation of the chemical and biological stages for the radiolysis in water [63,66] and associated rate constants.

Reactions in Water			
No.	Reaction	k (M^−1^ s^−1^)	Reference
1	e_aq_^−^ + e_aq_^−^ → H_2_ + 2OH^−^	6.47 × 10^9^	[63]
2	H• + H• → H_2_	5.03 × 10^9^	[63]
3	•OH + •OH → H_2_O_2_	4.75 × 10^9^	[63]
4	e_aq_^−^ + H• → H_2_ + OH^−^	2.65 × 10^10^	[63]
5	e_aq_^−^ + •OH → OH^−^	2.95 × 10^10^	[63]
6	H• + •OH → H_2_O	1.44 × 10^10^	[63]
7	e_aq_^−^+ H_2_O_2_ → •OH + OH^−^	1.41 × 10^10^	[63]
8	e_aq_^−^ + O_2_ → O_2_^−^	1.90 × 10^10^	[63]
9	e_aq_^−^ + O_2_^−^ → OH^−^ + HO_2_^−^	1.30 × 10^10^	[63]
10	e_aq_^−^ + HO_2_•→ HO_2_^−^	2.00 × 10^10^	[63]
11	H• + H_2_O_2_ → •OH + H_2_O	0.01 × 10^10^	[63]
12	H• + O_2_ → HO_2_•	2.00 × 10^10^	[63]
13	H• + HO_2_•→ H_2_O_2_	2.00 × 10^10^	[63]
14	H• + O_2_^−^ → HO_2_^−^	2.00 × 10^10^	[63]
15	•OH + H_2_O_2_ → HO_2_•+ H_2_O	2.30 × 10^07^	[63]
16	•OH + O_2_^−^ → O_2_ + OH^−^	9.00 × 10^09^	[63]
17	•OH + HO_2_• → O_2_ + H_2_O	1.00 × 10^10^	[63]
18	HO_2_• + HO_2_• → H_2_O_2_ + O_2_	7.60 × 10^5^	[63]
19	HO_2_• + O_2_^−^ → H_2_O_2_ + O_2_ + OH^−^	1.14 × 10^8^	[66]
20	H^+^ + OH^−^ → H_2_O	1.00 × 10^11^	[63]
21	H_2_O → H^+^ + OH^−^	1.29 × 10^−5^	[66]
22	H^+^ + HO_2_^−^ → H_2_O_2_	2.00 × 10^10^	[63]
23	H_2_O_2_ → H^+^ + HO_2_^−^	6.49 × 10^−2^	[66]
24	H_2_O_2_ + OH^−^ → HO_2_^−^+ H_2_O	1.18 × 10^10^	[66]
25	HO_2_^−^+ H_2_O → H_2_O_2_ + OH^−^	9.97 × 10^5^	[66]
26	•OH → H^+^ + O^−^	6.49 × 10^−2^	[66]
27	H^+^ + O^−^ → •OH	4.52 × 10^10^	[66]
28	•OH + OH^−^ → O^−^ + H_2_O	1.18 × 10^10^	[66]
29	O^−^ + H_2_O → •OH + OH^−^	9.97 × 10^5^	[66]
30	HO_2_•→ H^+^ + O_2_^−^	6.62 × 10^5^	[66]
31	H^+^ + O_2_^−^ → HO_2_•	3.00 × 10^10^	[63]
32	HO_2_•+ OH^−^ → O_2_^−^ + H_2_O	9.78 × 10^−2^	[66]
33	O_2_^−^ + H_2_O → HO_2_• + OH^−^	1.18 × 10^10^	[66]
34	H^+^ + e^−^_aq_ → H•	2.11 × 10^10^	[63]
35	H• → H^+^ + e^−^_aq_	3.7	[66]
36	H• + OH^−^→ e^−^_aq_+ H_2_O	2.00 × 10^7^	[63]
37	e^−^_aq_+ H_2_O → H• + OH^−^	1.23 × 10^1^	[66]
38	H• + H_2_O → H_2_ + •OH	2.26 × 10^−5^	[66]
39	H_2_ + •OH → H• + H_2_O	4.50 × 10^7^	[63]

**Table 6 ijms-23-13484-t006:** Reaction set used in the simulation of the chemical and biological stages for the radiolysis in biological media [32] and associated rate constants. Concentrations for the different elements considered in the cell media are [RH] = 1 M, [GSH] = 6.5 mM, [Fe^2+^] = 0.89 μM, [lipid] = 1 μM, [catalase] = 0.08 μM [32]. Reactions whose suppression in the simulation contributes more than 1% to the relative change in the final AUC-ROO are highlighted in light red.

Reactions in Biological Media			
No.	Reaction	k (M^−1^ s^−1^)	Reference
40	RH + e^−^_aq_ → RH^−^	1.40 × 10^8^	[32]
41	RH + H• → RH (+H)	1.00 × 10^8^	[32]
42	2H^+^ + 2O_2_^−^ → H_2_O_2_ + O_2_	2.00 × 10^9^	[32]
43	2H^+^ + 2O_2_^−^ → H_2_O_2_ + O_2_	2.00 × 10^5^	[32]
44	GSH + •OH → H_2_O + GS	1.00 × 10^10^	[32]
45	•OH + RH → H_2_O + R•	1.00 × 10^9^	[32]
46	R• + O_2_ → ROO•	5.00 × 10^7^	[32]
47	R• + GSH → RH + GS	3.00 × 10^2^	[32]
48	R•+ R•	5.00 × 10^7^	[32]
49	ROO• + XSH → ROOH + XS	4.08 × 10^−2^	[32]
50	ROO•+ lipid → ROOH + R•	2.00 × 10^1^	[32]
51	2ROO• → O_2_ + ROH + RO•	1.00 × 10^4^	[32]
52	Fe^2+^ + H_2_O_2_ → Fe^3+^ + OH^−^ + •OH	1.00 × 10^3^	[32]
53	2H_2_O_2_ + catalase → O_2_ + 2H_2_O	6.62 × 10^7^	[32]

**Table 7 ijms-23-13484-t007:** Diffusion coefficients used in the simulations [63].

Chemical Species	Diffusion Coefficient (10^−9^ m^2^ s^−1^)
e^−^_aq_	4.9
H•	7.0
H^+^	9.46
H_2_	4.8
•OH	2.2
OH^−^	5.3
H_2_O_2_	2.3
HO_2_•	2.0
HO_2_^−^	2.0.
O_2_	2.1
O_2_^−^	2.1

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
