# Peer review of "Radical Production with Pulsed Beams: Understanding the Transition to FLASH"

_ijms, 2022, doi:10.3390/ijms232113484_

Round 1

Reviewer 1 Report

The Authors present a methodological study on the influence of different pulse dose rates on the biological effects that can affect the underlying mechanism of FLASH radiotherapy by simulating the chemical stage of water radiolysis with an in-house code developed for this purpose. The work is interesting and well presented. Before recommending the paper for publication, I suggest to the authors to address some general comments that I provide below.

Authors should give a short description/explanation for the non-familiar readers to the TOPAS-nbio and Geant4-DNA toolkits. Also a recent reference to Geant4-DNA on page 15 of the methodology like Kyriakou et al., Cancers 14,35 (2022) that also covers chemical and radiobiological aspects of the code, could be given. Moreover, although it is not the main purpose, nothing is said in the paper about how the physical stage could affect the reliability of the results.

I think a more detailed description of the simulation methodology would make the work more complete. Also, some more conclusive arguments in the discussion section would strengthen the reliability and value of the work.

Additionally, since the Authors employ MC simulations of the chemical stage, they could further discuss the relevance of their treatment compared to the works of Ramos-Mendez et al., Medical Physics 47, 5919 (2020) , Ref. 30 of this manuscript and also Tran et al., Medical Physics 48, 890 (2021). In this context, I think that the introduction although very well written, does not cover adequately the field from the radiochemistry stage point of view.

On page 14, please explain for the readers what do you mean by “ionization cross-sections on various molecular compartments”.

Reviewer 2 Report

A good work was implemented and discussed using MC simulation of Radical production under FLASH therapy by GPU code. However, some general remarks are needed :-

> In Table 5 and 6, please put bracket (i.e [50] in last column as it refer to References

> Could you define the constant fc in production term of  Eq. 7   1.602 * 10–17

> please add (YES) in 3D section worlflow of figure 9 

>could you please write in discussion section about the  small differences appeared in the fitting parameters and present it in percentage number. Also, write the relates references that discuss the   heterogeneous nature of  chemical phase

> Conclusion  section need to be paraphrased as it should include the important scientific number in my opinion, avoid using  word "we" in conclusion part.  

Reviewer 3 Report

t     The study by A. Espinosa Rodriguez et al. analyzes an important anticancer treatment modality such as FLASH radiotherapy. This new and exciting approach deserves an in-depth multidisciplinary investigation. The study presents a series of simulations of generation of free radicals and the outcome of this key process in irradiated tissues. 

       A few questions before the study can be recommended for publication: 

      1. Please explain what is Np in the formula (4).

2.      2. Sometines (e.g., Fig. 2, Table 3), the results are shown without deviations. Please indicate the statistically significant differences where it is meaningful or explain the range of errors.

3.      3. The concentrations of radicals are calculated based on the solutions of the system of differential equations with many parameters. Did the authors study the reliability of decisions regarding the values of these parameters? What is the error of the numerical solution and the net error considering the errors between the coefficients?

4.      4. What temperature corresponds to the constants? How is the local non-homogeneity of temperature taken into account during irradiation with different pulse durations?

5.      5. Fig. 10: does the logistic fit really describe the experimental data?

Round 2

Reviewer 3 Report

The authors addressed the comments and added valuable fragments. Now the study can be accepted for publication.